# Elliptical Bloch skyrmion chiral twins in an antiskyrmion system

Jagannath Jena[1,4], Börge Göbel [1,2,4], Tianping Ma[1], Vivek Kumar [3], Rana Saha [1], Ingrid Mertig[2,1], Claudia Felser [3] & Stuart S.P. Parkin [1✉]

Skyrmions and antiskyrmions are distinct topological chiral spin textures that have been observed in various material systems depending on the symmetry of the crystal structure. Here we show, using Lorentz transmission electron microscopy, that arrays of skyrmions can be stabilized in a tetragonal inverse Heusler with $D_{2d}$ symmetry whose Dzyaloshinskii-Moriya interaction (DMI) otherwise supports antiskyrmions. These skyrmions can be distinguished from those previously found in several B20 systems which have only one chirality and are circular in shape. We find Bloch-type elliptical skyrmions with opposite chiralities whose major axis is oriented along two specific crystal directions: [010] and [100]. These structures are metastable over a wide temperature range and we show that they are stabilized by long-range dipole-dipole interactions. The possibility of forming two distinct chiral spin textures with opposite topological charges of ±1 in one material makes the family of $D_{2d}$ materials exceptional.

[1] Max Planck Institute of Microstructure Physics, Weinberg 2, 06120 Halle (Saale), Germany. [2] Institute of Physics, Martin Luther University Halle-Wittenberg, 06120 Halle (Saale), Germany. [3] Max Planck Institute for Chemical Physics of Solids, Nöthnitzer Strasse 40, 01187 Dresden, Germany. [4]These authors contributed equally: Jagannath Jena, Börge Göbel. ✉email: stuart.parkin@mpi-halle.mpg.de

Non-collinear spin textures range in size from atomic scale to microscopic and have attracted much interest because of their unique physical properties and potential applications in spintronic devices such as the racetrack memory[1–18]. Of these spin textures, skyrmions and antiskyrmions are particularly fascinating because they are magnetic objects with distinct chiral magnetic boundaries. Both are characterized by opposite topological charges of ±1 and have only been predicted to coexist in non-chiral systems like in frustrated magnets[19,20]. However, experimentally, this coexistence has not been observed, yet. Which object is actually found in a given magnetic material depends on the underlying symmetries of its crystal structure determining the tensor of the chiral Dzyaloshinskii–Moriya interaction (DMI)[21,22]. For example, perhaps the most extensively studied systems are the chiral cubic B20 systems, such as MnSi[2] and FeGe[6], whose symmetry gives rise to skyrmions, which are circular with Bloch boundaries. On the other hand, antiskyrmions have only been observed, to date, in $Mn_{1.4}Pt_{0.9}Pd_{0.1}Sn$. These objects have boundaries that are segmented into alternating Bloch and Néel regions, with opposite chiralities. These segments are fixed to the underlying crystal structure, which has $D_{2d}$ symmetry.

Here we show that in this same system, these antiskyrmions can be transformed into a new type of skyrmion, an elliptical Bloch skyrmion, when a different experimental protocol is followed. Individual objects and arrays of elliptical skyrmions can be formed in zero magnetic field over a wide range of temperature. We show, using micromagnetic simulations, that long-range dipole–dipole interactions allow for the formation of these metastable objects that have Bloch walls. We observe two types of elliptical skyrmions with opposite chirality and different elongation directions, which are tied to the crystal's high symmetry directions. The ellipticity is therefore an intrinsic property of materials with $D_{2d}$ symmetry, in contrast to the recently observed distorted skyrmions in confined B20 materials[23–28]. The formation of two distinct chiral spin textures with opposite topological charges of ±1 in one material system makes the $D_{2d}$ and related systems especially fascinating.

## Results

**Experimental details**. Our studies here concern the tetragonal Heusler compound $Mn_{1.4}Pt_{0.9}Pd_{0.1}Sn$ that was prepared in the form of polycrystals with single crystal grains of typical size ~60–80 μm, as in our previous work[14]. Thin lamellae, with a thickness of ~170 nm, were fabricated from one single crystalline grain oriented along [0 0 1] using $Ga^+$ focused ion beam (FIB) techniques. The crystal orientation was determined from in situ electron backscattering diffraction (EBSD). Details of the complete procedure are given in the Supplementary Fig. 1. Lorentz transmission electron microscopy (LTEM) was used to image the magnetic textures. The lamella is placed in a double tilt, variable temperatures, sample stage. Except where noted, all the images presented here are from the same 1200 nm × 1200 nm region of the lamella. Similar images were found in other regions and lamellae. A magnetic field, which is oriented along the TEM column, is applied to the sample by passing current through the objective lens. By tilting the sample, a component of the magnetic field in the plane of the sample can be applied.

**Stabilization of antiskyrmions**. The formation of antiskyrmions depends strongly on the field-temperature history. We find, as in our earlier work, that the use of an in-plane magnetic field within the basal plane of the inverse tetragonal structure of $Mn_{1.4}Pt_{0.9}Pd_{0.1}Sn$ is essential to stabilize antiskyrmions[14], since an energy barrier between the topologically trivial and non-trivial states needs to be overcome. This is illustrated in Fig. 1. We apply a strong out-of-plane magnetic field that is just sufficient to bring the system into a fully polarized state at a given temperature (Fig. 1a). From there we gradually decrease the field and observe the emergence of the helical phase (Fig. 1b–f). No antiskyrmions are formed here where the in-plane component of the field has been set to zero[14], since the energy barrier between the helical phase and the antiskyrmion phase is apparently too large. Next, we include an in-plane component via sample tilting using the following experimental protocol (schematically shown in Fig. 1m, n): from the field-polarized state, we reduce the field to a particular value and then tilt the sample by 35°–40° away from the normal direction to approximately along the [1 1 0] direction to help to overcome the energy barrier. The sample is then tilted back to 0° and an LTEM image is acquired. The field is then decreased in steps. At each field LTEM images are collected after the sample is tilted away from and back to zero. This procedure is carried out until a periodic lattice of non-collinear objects has formed. From this point on we simply decrease the field without tilting, since we want to investigate the topologically non-trivial phase and we do not want to overcome the barrier to the helical phase. Results using this protocol are shown in Fig. 1g–l at 300 K. A few circular features emerge at 368 mT (Fig. 1g) and a densely packed array of these objects is found for fields between 336 and 288 mT (Fig. 1h). These features are consistent with anti-skyrmions[14]. The LTEM contrast is symmetric showing two spots of increased intensity and two spots of reduced intensity coming from the oppositely oriented Bloch parts of the antiskyrmion. From here we decrease the field without tilting and observe a transformation of the round antiskyrmions to square-shaped features in the LTEM contrast (Fig. 1i). In this panel intermediate states are observable. For lower fields only square-shaped features occur and form a lattice (224 mT in Fig. 1j), which is stable even at zero field (Fig. 1k). Finally, at −128 mT the lattice turns into a helical phase with helix orientations along the [1 0 0] and [0 1 0] directions (Fig. 1l). The square-shaped contrast shows a black surrounding along the [0 1 0] and [0 $\bar{1}$ 0] directions, and a white surrounding along the [1 0 0] and [$\bar{1}$ 0 0] directions, similar to the before described antiskyrmion, which indicates that both objects are topologically equivalent. We will elaborate on the reason of the shape deformation later in this paper by micromagnetic simulations.

The role of the in-plane field component for the formation of antiskyrmions becomes also apparent when we stop tilting the sample already at a higher magnetic field, for which a dense antiskyrmion lattice is not yet seen (cf. Supplementary Fig. 2). The sparse array of antiskyrmions does not survive when the field is decreased without any tilting. At zero field, the texture shows only helices and no antiskyrmions.

At 350 K (Fig. 2) we also observe square-shaped objects, but the field dependence is distinct as compared to 300 K. Starting from the field-polarized phase, we observe isolated features that are not circular (Fig. 2a). When further decreasing the field, a periodic lattice of square antiskyrmion-like objects forms (Fig. 2b). These transition to a helical phase between 96 and 0 mT where the number of individual objects gradually is reduced (Fig. 2c–e). We observe no round antiskyrmions at this temperature.

**Elliptical skyrmions at 200 K**. Next, we present results at a lower temperature of 200 K (Fig. 3). Starting once again from the field-polarized phase, as the field is reduced, we find an entirely new type of magnetic nano-structure, namely nano-objects that are elliptical rather than circular or square in shape. These new elliptical structures appear first as isolated objects (Fig. 3a at 432 mT) and at lower fields as hexagonal lattices (Fig. 3b for 208 mT). The ellipses are characterized by a long axis that is oriented along

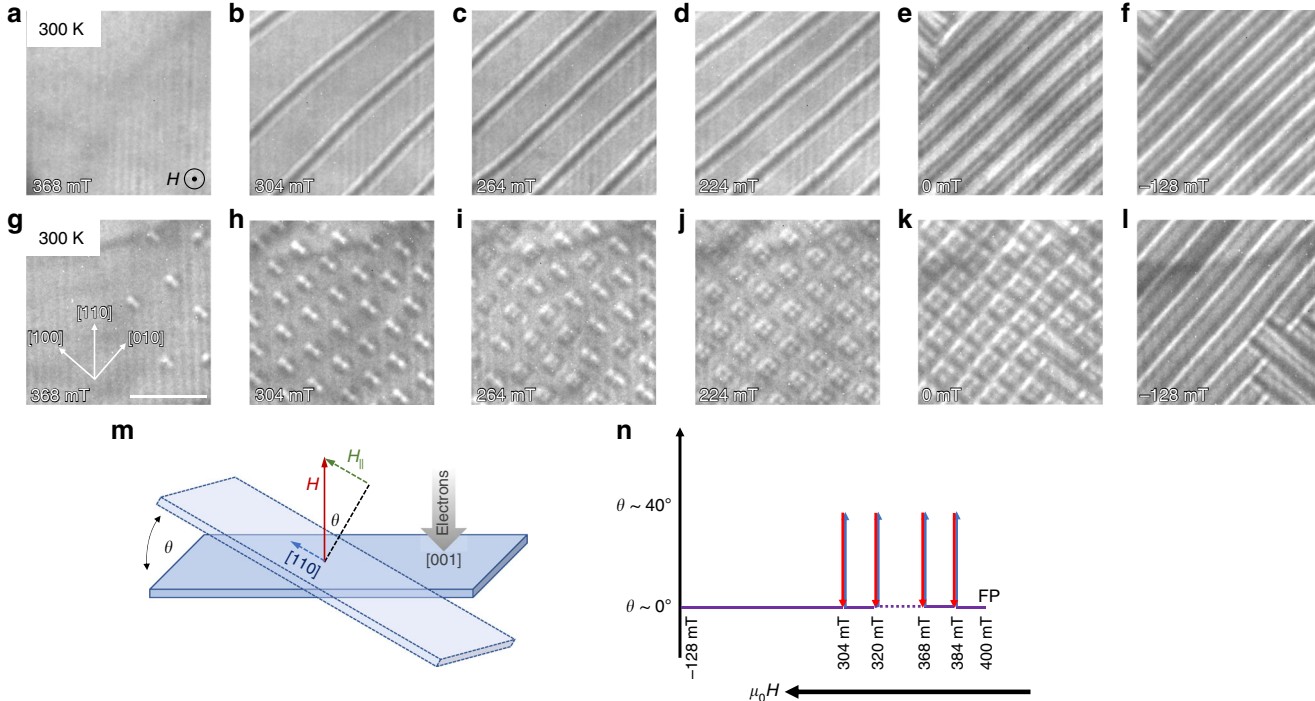

**Fig. 1 Field-induced stabilization of antiskyrmions in $D_{2d}$ material.** The top row (**a**–**f**) shows the observation under the decreasing field mode at 300 K. A magnetic field is applied and decreased coming from the field-polarized phase. The texture exhibits helices with decreasing period length. In the bottom row (**g**–**l**) we apply the modified protocol (see schematic figures **m**, **n**) where we tilt the field B by 35°–40° in the [1 1 0] direction and come back to [0 0 1] orientation (as indicated) at high fields until a dense array of antiskyrmions has formed in **h**. We observe two different types of features: round ones indicating conventional antiskyrmions and square ones, which are topologically identical to antiskyrmions. A highly periodic lattice of the square objects is metastable even at vanishing field (**k**) and transforms into the helical phase only at a negative field of −128 mT. The scale bar corresponds to 500 nm. In **m** we show the tilting of the sample and **n** illustrates the procedure that was used for the measurement of the LTEM images shown in **g**–**l**.

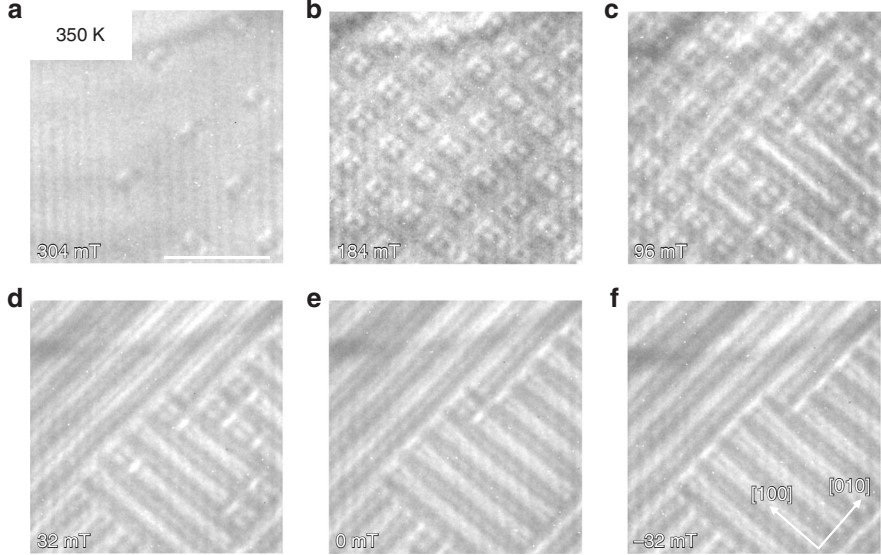

**Fig. 2 Formation of square antiskyrmions.** The figure shows the evolution of the LTEM contrast with decreasing field applying the modified experimental protocol (applying reversible tilting along [1 1 0] at high fields until a lattice of objects has formed) at the highest investigated temperature of 350 K. We observe the transition from the field-polarized state, to a few isolated objects in **a**, to a lattice of square antiskyrmionic features in **b**, **c**, to the helical phase in **d**–**f**. The scale bar corresponds to 500 nm.

[1 0 0] and a short axis that is directed along [0 1 0]. These objects in the LTEM images are surrounded by a black line with a white contrast in the center at higher fields. Upon decreasing the field, these objects become larger and a gray contrast appears in their center. The objects exist at zero field (Fig. 3e) and are partially metastable even for fields as large as −48 mT, where a transition to a helical phase sets in (Fig. 3f). These objects would appear to look like Bloch-like skyrmions, as previously observed[2,3,6], but these

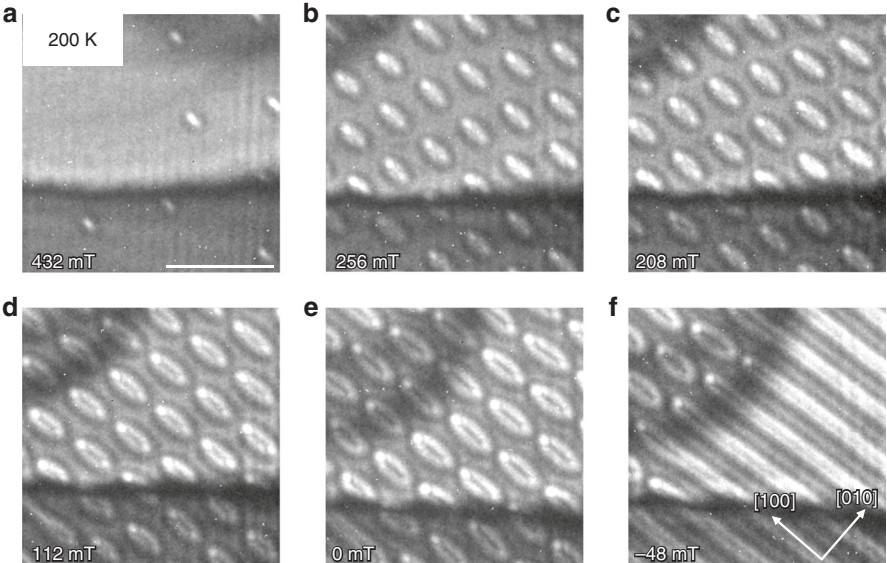

**Fig. 3 Formation of elliptical skyrmions.** The figure shows the evolution of the LTEM contrast with decreasing field applying the modified experimental protocol (applying reversible tilting along [1 1 0] at high fields until a lattice of objects has formed) at the temperature of 200 K. Elliptic objects are observed that are identified as elongated Bloch skyrmions. One observes a few skyrmions at high fields (**a**) and then a transition to a densely packed array of skyrmions (**b–e**), which transforms into the helical phase only at negative fields (**f**). The black horizontal line in each LTEM image corresponds to bending contours. The scale bar corresponds to 500 nm.

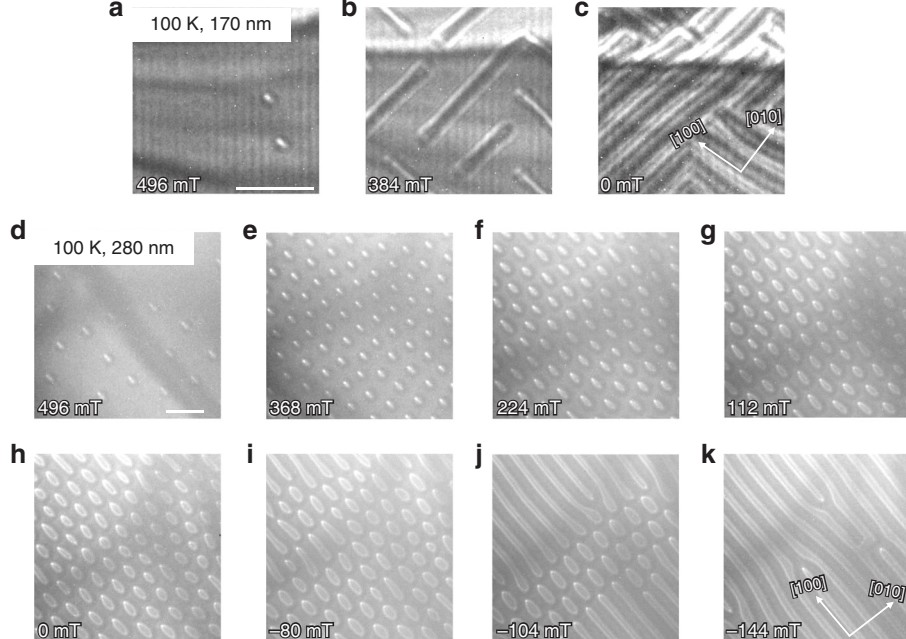

**Fig. 4 Elliptical skyrmions at 100 K for two different sample thicknesses.** In **a–c** LTEM measurements in the previously investigated sample (170 nm thick) are shown. The field is reduced coming from the ferromagnetic phase, with only a few skyrmions shown in **a**. In **b** helices have formed that constitute the ground state at zero field in **c**. In **d–k** the sample has a thickness of 280 nm. Starting again at high fields (**d**), a dense array of elliptical skyrmions forms (**e**), which persists upon decreasing the field (**f, g**), even down to zero field (**h**) and below (**i**). Finally, at −104 mT (**j**), the helical phase starts to dominate. At −144 mT in **k**, the helical ground state is restored. The scale bars correspond to 500 nm.

"conventional" skyrmions have been found to be circular and not elliptical in shape as we observe here.

Note that a similar behavior is also found at 250 and 150 K (cf. Supplementary Fig. 3). At 100 K, we only observe a few isolated elliptical skyrmions that directly transition to the helical phase (see Fig. 4a–c). However, in a thicker sample, a periodic lattice of elliptical skyrmions has been stabilized even

at such low temperatures (Fig. 4d–k) pointing towards the relevance of dipole–dipole interactions in this material.

The $D_{2d}$ symmetry of the tetragonal Heusler system gives rise to an anisotropic DMI[21,22], which strictly favors the stabilization of antiskyrmions over that of conventional skyrmions[14,29]. In order to account for the unusual elliptically shaped nano-objects that we see at low temperatures in $Mn_{1.4}Pt_{0.9}Pd_{0.1}Sn$, we have

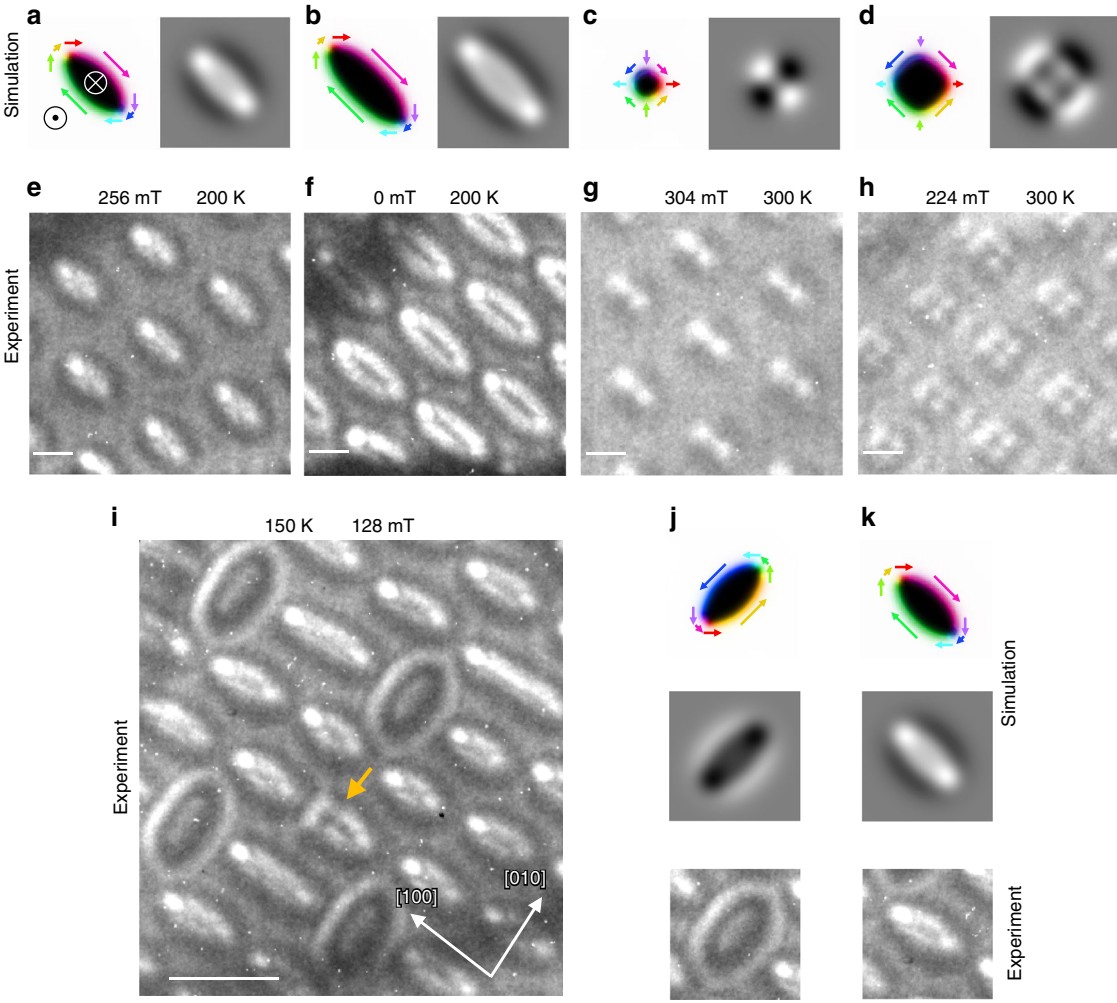

**Fig. 5 Micromagnetic simulations of elliptical skyrmions and antiskyrmions.** In **a–d** different magnetic textures are stabilized starting from a skyrmion or an antiskyrmion. The resulting textures are elongated skyrmions in **a**, **b**, a round antiskyrmion in **c**, and a square-shaped antiskyrmion in **d**. In **a**, **b** we started from a Bloch skyrmion configuration and in **c**, **d** from an antiskyrmion. The out-of-plane magnetic fields in the simulations are: **a** 180 mT, **b** 170 mT, **c** 210 mT, and **d** 180 mT. The calculated LTEM images (right panels) are in good agreement with the experimentally observed patterns (**e–h**). The magnified images in **e–h** are taken from Fig. 3b (256 mT, 200 K), 3e (0 mT, 200 K), 1h (304 mT, 300 K), and 1j (224 mT, 300 K), respectively. A comparison of the experimentally observed lattices of round and square antiskyrmions, as well as elliptical skyrmions, is shown in Supplementary Fig. 6. The scale bars for the LTEM images in **e–h** correspond to 100 nm. In **i–k** the coexistence of elliptical skyrmions with opposite chiralities is shown at 150 K and 128 mT. Micromagnetic simulations of the two objects (**j**, **k**) confirm this interpretation. Furthermore, a topologically trivial object is highlighted by the orange arrow. Here, the scale bars correspond to 300 nm.

carried out micromagnetic simulations (see Methods) for this material that include a DMI interaction consistent with the $D_{2d}$ symmetry, and take into account dipole–dipole interactions.

**Explanation and micromagnetic simulations**. To explore possible metastable magnetic configurations, we carry out micromagnetic simulations that are initialized with circularly shaped, topologically different magnetic states. The process of relaxation is shown in Supplementary Fig. 4; in the following, we discuss the relaxed, metastable configurations (labeled as "simulation" in Fig. 5a–d).

When the initial state is set to be an antiskyrmion, we find, after relaxation, that this object is indeed a metastable configuration for the magnetic parameters chosen (Fig. 5c). When the initial state is set to be a particular Bloch skyrmion with a given chirality (here clockwise), we observe that the object converges to an object with an elliptical shape, as we find experimentally (Fig. 5a). Note that, similar to the protocol dependence of the

experiments, the metastabilized states in the simulations depend on the initial magnetic texture. This means, for example, that not every initial Bloch skyrmion configuration leads to the stabilization of a metastable object. However, if a nano-object is successfully stabilized, it is an elliptically elongated Bloch skyrmion. A very interesting point that we find from these simulations is that the chirality of the initial object determines along which direction the major axis of the ellipse is oriented. The ellipses are always oriented with their major axis along one of two perpendicular in-plane directions ([1 0 0] or [0 1 0]).

The elliptical objects are Bloch like because of dipole–dipole interactions, since costly volume charges div(**m**) are absent in Bloch walls[30]. The dipole–dipole interactions are of relevance in this system since the sample thickness and the saturation magnetization are considerable. As mentioned above, one of the relaxed Bloch skyrmions is elongated along [1 0 0] (Fig. 5a, b) as in our experiments, which we attribute to the anisotropic DMI. This type of DMI favors Néel configurations along the in-plane {1 1} crystallographic directions and Bloch configurations along

the in-plane {1 0} directions. Importantly, the favored chirality of the Bloch parts differs locally: a clockwise state is favored along [0 1 0] and [0 $\bar{1}$ 0] (green and pink arrows), while counter-clockwise parts (blue and yellow) are favored along [1 0 0] and [$\bar{1}$ 0 0]. Consequently, the favorable parts of the Bloch skyrmion are enlarged, effectively elongating the skyrmion along the [1 0 0] direction here. The simulated LTEM images for such a configuration (Fig. 5a, b) well describe the experimental observations even at different magnetic fields strengths (Fig. 5e, f). Simulated LTEM images for different defocus values are shown in Supplementary Fig. 5. When the initial Bloch skyrmion is chosen to have a counter-clockwise chirality (cf. Supplementary Fig. 4c), the relaxed state is again an elliptical object, but which is now elongated along the perpendicular [0 1 0] direction (see Fig. 5j). This time the blue and yellow highlighted Bloch parts are favored by the anisotropic DMI and are consequently enlarged.

We have indeed observed experimentally that the elliptical skyrmions are oriented along just two crystal directions, and indeed those that our micromagnetic simulations showed were energetically favored (Fig. 5i–k). Moreover, the LTEM images show that the elliptical skyrmions in $Mn_{1.4}Pt_{0.9}Pd_{0.1}Sn$ have specific and opposite chiralities for these two crystal directions. Thus, these elliptical skyrmions come as "chiral twins."

This makes elliptical skyrmions fundamentally different from DMI-stabilized skyrmions in B20 materials, where only one chirality is allowed, determined by the chirality of the material[3,6,21,22]. There is no support, either experimentally or theoretically for such elliptical skyrmions in extended materials. For example, micromagnetic simulations in which an isotropic DMI, reflective of the cubic B20 symmetry is used, show circularly shaped Bloch skyrmions, even after taking into account dipole–dipole interactions[8,31]. Note that there exist reports of deformed or distorted skyrmions in several B20 materials[24–28]. However, the deformations in these cases are irregular, and are either a result of anisotropic $q$-vectors[24] or of confinement in nano-structured materials, from nano-sized grains or laterally patterned nano-structures. These deformations are not tied to the underlying crystal structure and, thus, are not an intrinsic property as we find here.

The influence of dipole–dipole interactions also becomes apparent in the micromagnetic simulations of an initial antiskyrmion after relaxation to a metastable configuration (Fig. 5c, d). Interestingly, here we find two distinct shapes after relaxation. In larger magnetic fields, where the object is smaller, the original circular shape is maintained but, in lower magnetic fields, a square deformation takes place: the Bloch parts of the antiskyrmion along the family of in-plane {1 0} directions (pink, blue, green, and yellow arrows) are favored by the dipole–dipole interactions, leading to an enlargement of these parts. On the contrary, the Néel parts are unfavorable, leading to a spatial shrinking[32]. The result is a square-shaped antiskyrmion with a calculated LTEM signal (see Methods) similar to the experimental observations (cf. Fig. 5d, h). In higher magnetic fields, the antiskyrmion decreases in size (Fig. 5c) so that dipole–dipole interactions become less important due to the shortened wall lengths. Since the domain wall width is now comparable to the size of the magnetic object, Bloch and Néel parts become almost energetically equivalent and the square deformation is not noticeable anymore[32].

In this regard, we note that there is a clearly noticeable transformation process between them as observed in Fig. 1h–j. In Supplementary Fig. 7, we show that round antiskyrmions can transform to square-shaped antiskyrmions when the magnetic field is simply decreased. However, when the field is increased again, the state remains square shaped. The transition to round antiskyrmions is only possible when an in-plane field is temporarily

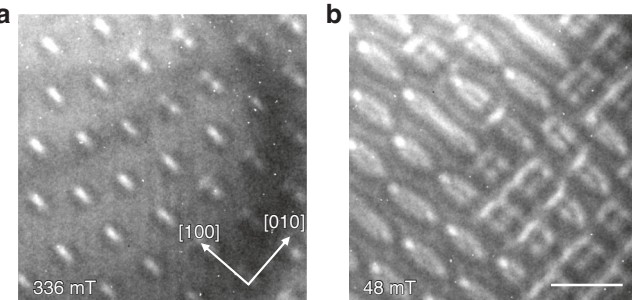

**Fig. 6 Coexistence of antiskyrmions and elliptical skyrmions.** Near a transition temperature of 268 K, we find both topologically distinct nano-objects simultaneously. **a** shows the results at 336 mT and **b** at 48 mT. In agreement with the measurements at 300 K (Fig. 1), we observe round antiskyrmions at elevated magnetic field strengths (336 mT) and square antiskyrmions at low fields (48 mT). At both field strengths elliptical skyrmions are present, whose size increases when the field is decreased. The scale bar corresponds to 300 nm.

provided. There appears to be a small energy barrier between the round and square antiskyrmion lattices in our experiments. In the case of isolated objects in an ideal sample, as presented in the micromagnetic simulations, this barrier is absent. For this reason, the interaction between neighboring objects also seems to have an influence on the metastability of magnetic objects in $D_{2d}$ materials. The possibility for controlled switching between both states is promising for future applications.

Since we observed antiskyrmions near room temperature and elliptical skyrmions at 250 K and below, we also investigated the critical behavior at intermediate temperatures. Near 268 K we find coexisting skyrmions and antiskyrmion (round at high fields in Fig. 6a and square shaped at low fields in Fig. 6b), indicating that both nano-objects have a comparable energy at this temperature with a presumably decreased energy barrier.

## Discussion

The reported metastability of round- and square-shaped antiskyrmions, as well as elliptically deformed Bloch skyrmions with two different chiralities arises from the anisotropic DMI in Heusler materials compared to traditional skyrmion hosts. The anisotropic DMI favors the stabilization of antiskyrmions, while the dipole–dipole interactions favor Bloch-type skyrmionic objects. In Fig. 7 we show the phase diagram, which summarizes our results. Comparing this phase diagram with our initial publication[14] shows how important the experimental protocol is for the stabilized magnetic phase. In the present protocol, we find elliptical skyrmions at low temperatures (250 K and below) and round- and square-shaped antiskyrmions near and above room temperature (see the phase diagram in Fig. 7). At a critical temperature near 268 K, we observe the coexistence of antiskyrmions and elliptical skyrmions (Fig. 6).

We relate the temperature dependence to the decreased net magnetization at higher temperatures, which reduces the importance of dipole–dipole interactions. To further investigate the role of the magnetization on the stability of elliptical skyrmions, we have measured a thicker sample (Fig. 4). While in the previously investigated sample (thickness 170 nm), only a very few individual skyrmions could be stabilized at 100 K (Fig. 4a–c), in the thicker sample (280 nm thickness), we observe a well-ordered skyrmion lattice (Fig. 4d–k).

Besides the coexistence of skyrmions and antiskyrmions in a single material that we observe for the first time, we also observe two different elliptically deformed Bloch skyrmions with opposite

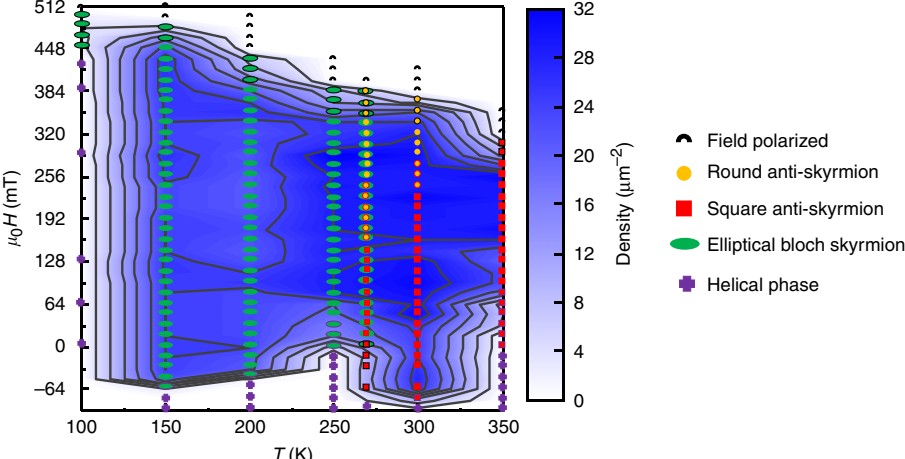

**Fig. 7 Magnetic phase diagram.** The observed magnetic textures form five categories: field-polarized state, round and square antiskyrmions, elliptical Bloch skyrmions and the helical phase, as indicated by the symbols shown in the figure. A mixed phase is shown by the corresponding symbol with an appropriately colored edge. The contour plot in the background shows the density of skyrmionics and antiskyrmionic objects. This phase diagram corresponds to a ~170-nm-thick sample with a tilted-field protocol as follows: at a given temperature the field is decreased, starting from the field-polarized phase, in steps of a few mT, then the sample is tilted ~40° away from the pole direction, and finally tilted back and the LTEM image is taken. Once a dense array of skyrmions or antiskyrmions is found, the field is subsequently decreased without any tilting. At 268 K we observe the coexistence of skyrmions and antiskyrmions (cf. Fig. 6).

chiralities. While our theoretical results predict equal probabilities for the formation of the two distinct elliptical skyrmions, in our experiments we mostly find Bloch skyrmions that are elongated along [1 0 0] (see Fig. 3) at temperatures between 150 and 250 K. Even when repeating the experiment with tilted magnetic fields along a different in-plane direction and tilting back, the result is the same (see Supplementary Fig. 8). This implies that the elongation direction is not much influenced by the direction of the in-plane field or the temperature in our case. However, in a different part of the sample (shown in Fig. 5i), we observe a mixed state of both skyrmion types. Therefore, we conclude that in the first case, the chirality of the skyrmions is fixed by the lattice itself. This could, for example, be caused by strain; even a small lattice deformation would lead to an inequivalence of the two elliptical-skyrmion chiralities. In the second part of the sample, the deformation could be negligible, which is why both chiralities would be allowed. Furthermore, it is worth mentioning that in this part of the sample, we even observe what appears to be a topologically trivial object; a bubble that is the combination of a skyrmion and an antiskyrmion (highlighted in Fig. 5i).

In summary, we have observed two topologically distinct phases in a material with $D_{2d}$ symmetry: elliptical skyrmions and antiskyrmions. The observed antiskyrmions come in two varieties, square and round shaped, which can be switched reversibly. We observed individual objects of all three types and observed lattices of elliptical skyrmions and square-shaped antiskyrmions over a wide range of temperature even in zero magnetic field. The elliptical skyrmions are stabilized by dipole–dipole interactions, which leads, via an interplay with the anisotropic DMI in this $D_{2d}$ system, to their elongation along the [1 0 0] or [0 1 0] directions, namely the propagation direction of the ground state helical phase. Moreover, the elliptical skyrmions possess either a left- or right-handed chirality, depending on their elongation direction. This makes elliptical skyrmions distinct from the rotational-symmetric DMI-stabilized Bloch skyrmions in B20 materials. In the latter case, dipole–dipole interactions reinforce the DMI to favor circularly shaped skyrmions, without the possibility of creating spin textures with distinct topologies, as we find here in a $D_{2d}$ material. Our observation of magnetic non-collinear chiral spin textures with opposite topological charges in

$Mn_{1.4}Pt_{0.9}Pd_{0.1}Sn$ show that $D_{2d}$ materials have greater functionalities as compared to, for example, B20 materials. One can anticipate that their magnetization dynamics may also show greater complexity with the eventual possibility of switching between different topological textures.

Note added: During the review process we became aware of ref. [33], in which the existence of skyrmions, antiskyrmions, and topologically trivial bubbles in an unspecified non-centrosymmetric Heusler magnet is shown. The antiskyrmion shows a square deformation and the skyrmion is elliptically deformed along a single direction.

## Methods

**Experiment**. The Heusler compound $Mn_{1.4}Pt_{0.9}Pd_{0.1}Sn$ is a bulk polycrystalline sample, which was prepared by the arc-melting method as described in an earlier report[14]. The single crystalline lamellae of [0 0 1] orientation were prepared in a $Ga^+$ ion contained dual beam system of FIB [FEI Nova Nanolab 600 SEM/FIB] operating at an accelerated voltage of 30 keV. To find [0 0 1]-oriented grains, an electron backscattering diffraction (Quantax, Bruker) experiment is performed in a TESCAN GAIA 3 system. Various magnetic textures are investigated in an aberration-corrected high resolution transmission electron microscope [FEI TITAN 80-300] operated at an acceleration voltage of 300 keV in LTEM mode. The magnetic field was applied along the electron beam direction of the TEM by partially exciting the objective lens. An in-plane component of the magnetic field was applied by tilting the lamellae a few degrees away from the zone axis [0 0 1] using a double tilt liquid nitrogen holder that allows a maximum tilting angle of up to ±40°. The under-focus value for the LTEM images is 1 mm. Periodic vertical lines in the LTEM images are non-magnetic contrasts caused by the FIB specimen preparation.

**Micromagnetic simulations**. For the micromagnetic simulations the Landau–Lifshitz–Gilbert equation[34,35] was propagated for every discretized magnetization cell (normalized moment **m**) according to:

$$\dot{\mathbf{m}} = -\gamma_e \mathbf{m} \times \mathbf{B}_{\text{eff}} + \alpha\, \mathbf{m} \times \dot{\mathbf{m}}.$$

Here, $\gamma_e$ is the gyromagnetic ratio of an electron, and $\alpha = 0.3$ is the Gilbert damping parameter. $\mathbf{B}_{\text{eff}} = -\frac{\delta F}{M_s \delta \mathbf{m}}$ is the effective magnetic field derived from the free energy density $F$, taking into account Heisenberg exchange, dipole–dipole interactions, magnetic anisotropy, external magnetic field, and the DMI. Tetragonal Heusler compounds with a $D_{2d}$ symmetry exhibit an "anisotropic DMI" as follows:

$$H_{\text{DMI}} = D\left( m_z \frac{\partial m_x}{\partial x} - m_x \frac{\partial m_z}{\partial x} - m_z \frac{\partial m_y}{\partial y} + m_y \frac{\partial m_z}{\partial y} \right).$$

In $Mn_{1.4}Pt_{0.9}Pd_{0.1}Sn$ this form of the DMI emerges due to the layered nature of the crystal structure where the Pt atoms break inversion symmetry of a magnetic Mn layer differently along the [1 0 0] and [0 1 0] directions.

We take the following parameters from ref. [14], where antiskyrmions were initially reported: saturation magnetization $M_S = 445$ kA/m and exchange interaction $A_{ex} = 1.2 \times 10^{-10}$ J/m. The anisotropic DMI is characterized by the constant $D$. In ref. [14] $D$ was assumed to be 0.006 J/m$^2$, which was an overestimate, since the dipole–dipole interaction was neglected. Thus, here we use $D = 0.003$ J/m$^2$. A magnetic field $B_z$ was applied along the predominant magnetization orientation ($+z$) and an easy-axis anisotropy along $z$ was considered with $K_u = 0.2$ MJ/m$^3$. A sample size of 1200 nm $\times$ 1200 nm $\times$ 175 nm was simulated at a cell size of 5 nm $\times$ 5 nm $\times$ 5 nm. The simulations were carried out using the GPU (graphics processing unit)-accelerated program Mumax3[36,37], which was modified to include an anisotropic DMI.

**Simulation of LTEM images**. For the calculation of the LTEM images, we use a first-order approximation considering only a single layer (average magnetization of all layers) and a deflection of the incoming electron beams by the Lorentz force. The incoming electrons at the position $(x',y')$ are modeled by a Gaussian smearing function with a smearing factor $a = 60$ nm. The transmitted electron flux at $(x,y)$ is proportional to:

$$\int \exp\{-[(x - d \cdot m_y(x',y') - x')^2 + (y + d \cdot m_x(x',y') - y')^2]/a^2\} \, dx' \, dy'.$$

Here, $d = 2$ nm characterizes the maximum deflection of electrons by the Lorentz force.

## Data availability
The data that support the findings of this study are available from the corresponding author upon request. The micromagnetic simulation code is available at https://github.com/mumax/3/releases/tag/v3.9.3. Anisotropic DMI has been included in this code. The modified version is available from B.G. upon reasonable request.

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

## Acknowledgements
We thank the funding from the European Research Council (ERC) under the European Union's Horizon 2020 research and innovation program (grant agreement SORBET No. 670166). We also acknowledge the Deutsche Forschungsgemeinschaft (DFG, German Research Foundation)—Project number 403505322. B.G. and I.M. acknowledge funding by the Deutsche Forschungsgemeinschaft (DFG, German Research Foundation) under SFB TRR 227.

## Author contributions
J.J. and S.S.P.P. conceived the project. J.J. prepared the lamella, performed the LTEM measurements, and analyzed the data. The bulk material was synthesized by V.K. and C.F. B.G. performed the micromagnetic simulations and the LTEM image calculations with input from I.M. T.M. and R.S. contributed to data analysis. J.J., B.G., and S.S.P.P. wrote the manuscript with substantial contributions from all authors. S.S.P.P. supervised the project. All authors discussed the results.

## Competing interests
The authors declare no competing interests.
