## [Peer Review File · Nature Communications]

Reviewers' comments:

Reviewer #1 (Remarks to the Author):

Review of "Discovery of elliptical-skyrmion chiral twins in an anti-skyrmion system" by Jagannath Jena et al. submitted to Nature Communications.
Manuscript ID: NCOMMS-19-30742

The manuscript reports the "discovery" of two topologically distinct skyrmion phases in a single material with D_{2d} crystal symmetry. Using Lorentz TEM technique, they observed elliptical-skyrmion chiral twins with the skyrmion number +1 and antiskyrmions with the skyrmion number -1 in a thin plate of $Mn_{1.4}Pt_{0.9}Pd_{0.1}Sn$ depending on temperatures and bias-field strengths. Recently, similar observations using a material with exactly the same compositions have been reported by a research group at CEMS, RIKEN (L. Peng *et al.*, Real-Space Observation of a Transformation from Antiskyrmion to Skyrmion by Lorentz TEM. *Microsc. Microanal.* **25 (Suppl 2)**, 1840-1841 (2019), see attached file). I guess they must have already submitted their related manuscript to a high-impact journal. However, as far as I know, they are focusing on the in-situ observations of the in-plane field-induced transformation between the two states, as well as the intermediate state with the skyrmion number of 0. I suppose the present manuscript is presenting experimental results from different point of views to be distinguished from the related work. As examples, the present manuscript discusses the *out-of-plane field-induced* transformation from circular antiskyrmions to square antiskyrmions as shown in Fig. 1h-1i. In addition, the manuscript reports the emergence of elliptical Bloch-type skyrmions *at lower temperatures* (250K-150K; Fig. 3). Figure 4a(v)&4a(vi) demonstrate the transformation between the smaller elliptical Bloch-type skyrmion and the larger elliptical Bloch-type skyrmion, while Figure 4a(vii)&4a(viii) demonstrate the transformation between the circular antiskyrmion and the square antiskyrmion. Even the coexistence of elliptical-skyrmion chiral twins possessing opposite chiralities is clearly visualized (Fig. 4b(i)). These results are closely related with the asymmetric Dzyaloshinskii-Moriya interactions, which is specific to the tetragonal Heusler compound with D_{2d} symmetry. The present work demonstrates the possibilities to control various topologically different spin textures in a single material, which is expected to be useful in future spintronic applications. Since the two works must have been done independently, I think that the present work is also worth publishing in a similarly high-impact journal.

As a conclusion, I basically recommend the manuscript for a publication in *Nature Communications*. However, the present manuscript must be brushed up further before the final acceptance. Also, I would like to confirm several points to deepen my comprehension of their manuscript. In the following, I raise several questions and recommendations, and point out necessary revisions.

Questions and Recommendations

(1) Since the in-situ transformation between antiskyrmion and elliptical-skyrmion induced by weak in-plane bias-field has been reported by L. Peng *et al.*, add a discussion on the role of in-plane bias-field at a proper section in the main text. What is the role of

in-plane bias-field in creating the elliptical-skyrmion lattice ? Does the direction of the in-plane bias-field correlate with their chirality (CW/CCW) ? etc.

(2) The most interesting point of the present thin plate system is that a variety of unique spin textures emerge depending on field-temperature history, which is very complicated to follow. So, I strongly recommend the authors to prepare a schematic phase diagram (B-T) of these spin textures as an additional Figure 5 or as Fig.4c, with short notes at the end of Discussion in the main text.

(3) As shown in Fig. 3, only CW elliptical-skyrmions were observed at 200K. By decreasing the temperature to 150K, coexistence of CCW and CW elliptical-skyrmions was observed as shown in Fig. 4b(i). Does it follow that only CCW elliptical-skyrmions are to be observed at lower temperatures than 150K ? If the authors have already done their experiment at 90K or lower, I recommend the authors to include such results in their revised manuscript.

Revisions

(1) Title: "Discovery of elliptical-skyrmion chiral twins in an anti-skyrmion system"

Avoid "discovery". Choose an optimum title to be distinguished from the related work.

(2) Abstract, line 3: single "skyrmions" and arrays of "skyrmions" can be stabilized

I could not find any evidence that a single skyrmion is stabilized in Figs. 1-4, whereas sparse arrays of isolated skyrmions are observed in Figs. 1g,2a,3a. In addition, "skyrmion" is a vague expression. "A Bloch-type skyrmion with the skyrmion number of +1" is more precise.

(3) Abstract, line 5: "Dzyaloshinskii-Moriya"

Dzyaloshinskii-Moriya

=

(4) Abstract, line 7: "Here,"

Delete "Here". "Here" is already used in line 3.

(5) Abstract, line 12: "they are stabilized by long-range magnetostatic interactions"

"long-range magnetostatic interaction" is used only once in abstract. They use long-range dipole-dipole interaction" several times in the main text. Use a unified expression throughout the manuscript to avoid confusions if there is no special reason.

(6) Page 2, Introduction:

Introduction is too short (256 words). Add more background to increase the volume conforming to the journal style.

(7) Page 6, line 8-10: "When the initial Bloch skyrmion is chosen to have a counterclockwise chirality,..."

As this is illustrated in Fig. 4b(ii), add a reference at a proper position in the sentence.

(8) Page 7, line 9: "Note that there exist reports of deformed or distorted skyrmions in several B20 materials."

Origin of the L-shaped deformation of skyrmions reported in ref. 20 is the anisotropic q-vectors favoring [100] and [010] directions in $\text{Co}_8\text{Zn}_8\text{Mn}_4$. Other deformations as reported in refs. 21-24 were induced by confinement as correctly described in the present main text. Reference 20 must be described separately from references 21-24.

(9) Page 8, Discussion:

Discussion is too short (1 paragraph). Add a short discussion on the phase diagram as a summary of the present work. Also, it would be interesting if they could add discussion on their additional experiments at a liquid nitrogen temperature (or even lower).

(10) Throughout the main text:

Describe the defocus values (distinguish underfocus or overfocus) and indicate the direction of out-of-plane bias-field by using a proper symbol (either a target mark or a crossed circle).

(11) Figure 1g,2a,3a:

Describe whether these sparse isolated skyrmions remain even at vanishing field in the main text. I would like to know whether only a dense periodic lattice remains even after the bias-field is removed.

(12) Figures 1-4:

For readers' convenience, display the temperature as well as the strength of bias-field explicitly in all figures.

(13) Figures 1,2,4(vii),4(viii)

Poor contrast. Enhance contrast to improve visibility.

(14) Figure 1g

Display an arrow to show the direction of "stabilizing in-plane field [110]".

(15) Figure 3a

I notice a periodic stripe running in the vertical direction. Add a note whether this is the non-magnetic contrast produced in FIB specimen preparation or magnetic origin.

(16) Figure 4a(i),4a(ii), and 4b(ii)

Add the corresponding initial (circular?) states used to calculate the relaxed (elliptical) states as shown in 4a(i), 4a(ii), and 4b(ii) above each figure.

(17) Figure 4b(i):

The figure contains an intermediate state with the skyrmion number of 0 (see Fig. 2b of the attached literature) in the near center of the field of view. Indicate this unique spin texture by an arrow and add a note in both main text and figure legend.

(18) Figure 4:

Mixed displays of simulations and experimental images in a figure are so confusing. Add proper labels (such as Sim. and Exp.) to help readers' comprehension. In addition, add temperatures and field strengths in appropriate positions of each figure.

(19) References, 29:

Complete the reference in accordance with the journal style. I could not check the relevance of this reference.

End of Review

Reviewer #2 (Remarks to the Author):

In this study, the authors continued their investigation at exploring the existence of novel spin textures that can be stabilized in the tetragonal inverse Heusler $\text{Mn}_{1.4}\text{Pt}_{0.9}\text{Pd}_{0.1}\text{Sn}$ with D_{2d} symmetry. In a previous work, they observed by Lorentz TEM for the first time the room temperature stabilization of a lattice of anti-skyrmions due to the presence of an anisotropic DM interaction in this material. Here, the authors unambiguously observe another topological phase at low temperature composed of elliptical skyrmions, in which the elongation direction determines the effective sense of rotation. In addition, at temperature higher than room temperature, they observe square-shaped skyrmions either isolated or in a lattice phase depending on the applied magnetic field. Whereas the presence as well as the shape of the elongated skyrmions is corroborated by some micromagnetic simulations highlighting the importance of the long range dipole-dipole interaction, it would have been interesting to discuss more in detail the transition between circular and square skyrmionic textures at room temperature. Similarly, the authors explain the presence of only square skyrmions at higher than room temperature by invoking a decreased net magnetization at higher temperature. I believe that the authors should try to confirm this hypothesis by complementary experimental measurements completed by additional micromagnetic simulations. Once done, I would recommend this work for publication in Nature Communications.

Reviewer #3 (Remarks to the Author):

In this manuscript, Jena et al. report existence of elliptically deformed skyrmions in a Heusler compound with D_{2d} symmetry, which was previously found to host anti-skyrmions, by using Lorentz transmission electron microscopy technique. They argue that these nanometer-scale spin textures are stabilized by long range magnetostatic interaction by performing micromagnetic simulation that takes into account both DMI and dipole-dipole interaction.

I think that this paper contains novel and very interesting findings that warrant publication in Nature Communication, but before the publication, several points described below should be addressed to further polish the paper.

- 1) Authors apply a unique procedure for field application. What is the exact role or microscopic mechanism of the field tilting? How the in-plane field stabilizes antiskyrmions at high temperatures (300 -350 K) and skyrmions at low temperatures (150 K-250 K)? This point should be discussed.
- 2) To me, Fig.4 (v) and Fig.4(vii) look very similar to each other. Resolution of Fig.4(vii) is not so good, and it may be possible that the objects are actually skyrmions, not antiskyrmions. Also, is the symmetry of the each texture compatible with the symmetry of hexagonal arrangements if the objects are anti-skyrmion?
- 3) In this paper, authors state that at high temperatures at 300 K and 350 K, they observe antiskyrmions while at low temperatures (150-250 K), elongated skyrmions are observed. Are these results consistent with the phase diagram (Fig.4e) of the former Nature paper? In that figure, the whole phase diagram is dominated by antiskyrmions (although denoted as "skx"), and no skyrmion region is observed. What is the difference between the present and previous results?

Reply to Referee #1

The manuscript reports the "discovery" of two topologically distinct skyrmion phases in a single material with D2d crystal symmetry. Using Lorentz TEM technique, they observed elliptical-skyrmion chiral twins with the skyrmion number +1 and antiskyrmions with the skyrmion number -1 in a thin plate of Mn_{1.4}Pt_{0.9}Pd_{0.1}Sn depending on temperatures and bias-field strengths. Recently, similar observations using a material with exactly the same compositions have been reported by a research group at CEMS, RIKEN (L. Peng et al., Real-Space Observation of a Transformation from Antiskyrmion to Skyrmion by Lorentz TEM. *Microsc. Microanal.* 25 (Suppl 2), 1840-1841 (2019), see attached file). I guess they must have already submitted their related manuscript to a high-impact journal. However, as far as I know, they are focusing on the in-situ observations of the in-plane field-induced transformation between the two states, as well as the intermediate state with the skyrmion number of 0. I suppose the present manuscript is presenting experimental results from different point of views to be distinguished from the related work. As examples, the present manuscript discusses the out-of-plane field-induced transformation from circular antiskyrmions to square antiskyrmions as shown in Fig. 1h-1i. In addition, the manuscript reports the emergence of elliptical Bloch-type skyrmions at lower temperatures (250K-150K; Fig. 3). Figure 4a(v)&4a(vi) demonstrate the transformation between the smaller elliptical Bloch-type skyrmion and the larger elliptical Bloch-type skyrmion, while Figure 4a(vii)&4a(viii) demonstrate the transformation between the circular antiskyrmion and the square antiskyrmion. Even the coexistence of elliptical-skyrmion chiral twins possessing opposite chiralities is clearly visualized (Fig. 4b(i)). These results are closely related with the asymmetric Dzyaloshinskii-Moriya interactions, which is specific to the tetragonal Heusler compound with D2d symmetry. The present work demonstrates the possibilities to control various topologically different spin textures in a single material, which is expected to be useful in future spintronic applications. Since the two works must have been done independently, I think that the present work is also worth publishing in a similarly high-impact journal. As a conclusion, I **basically recommend the manuscript for a publication in Nature Communications**. However, the present manuscript must be brushed up further before the final acceptance. Also, I would like to confirm several points to deepen my comprehension of their manuscript. In the following, I raise several questions and recommendations, and point out necessary revisions.

We thank the referee for his/her very thorough review and his/her helpful comments. We are very glad for his/her positive evaluation of our work.

We were not aware of the abstract of Peng et al. but we are happy to include a note to this abstract as being found during the refereeing of our paper. The 1-page abstract by Peng et

al. describes some very nice work but due to its short length the information given is very limited. In particular, only single magnetic objects are discussed and it is not discussed whether they form as part of an array. Results are shown presumably at a single temperature but the temperature is not given. The authors only show a single image of a square-shaped antiskyrmion, a trivial bubble, and an elliptically deformed skyrmion. Only in this regard is there some overlap with our paper.

One of our most important discoveries is that the elliptical skyrmions are oriented along two distinct directions derived from the anisotropic DMI interaction. This important result is not discussed in the referenced abstract. We list below other significant results presented in our original paper that are not mentioned in the Peng et al. abstract plus additional results included in our revised version of our manuscript:

- the stabilization of lattices of square-shaped antiskyrmions, even at zero field
- the stabilization of lattices of round antiskyrmions and even individual objects
- the stabilization of lattices of elliptical-skyrmions, even at zero field, with two distinct elongation axes and chiralities
- the coexistence of both types of elliptical skyrmions
- the field-tilting dependence of the stability of elliptical-skyrmions
- the switching between round and square-shaped antiskyrmions
- the coexistence of skyrmions and antiskyrmions
- the temperature, field and protocol dependence of their stability (phase diagram)
- higher skyrmion stability for thicker samples
- a physical explanation considering anisotropic DMI and dipole-dipole interactions
- micromagnetic simulations that reproduce the experimental findings in agreement with our explanation

We thank the referee for acknowledging the significance of our results and for recommending our paper for publication.

(1) Since the in-situ transformation between antiskyrmion and elliptical-skyrmion induced by weak in-plane bias-field has been reported by L. Peng et al., add a discussion on the role of in-plane bias-field at a proper section in the main text.

Since skyrmions and anti-skyrmions have a finite topological charge, there exists an energy barrier between these topological phases and the topologically trivial phase (helical ground state or the saturated ferromagnetic state). As we have shown, an in-plane field is necessary to stabilize both the anti-skyrmion and the elliptical skyrmion lattice in this system. Therefore, we conclude that this field component helps to overcome the energy barrier in order to reach the energetically preferred skyrmion or anti-skyrmion phase. If the field is absent, the system remains in the topologically trivial phase even though this phase may be energetically less preferable. We added several comments on pages 3 and 4 to motivate the experimental protocol that we use.

What is the role of in-plane bias-field in creating the elliptical-skyrmion lattice ? Does the direction of the in-plane bias-field correlate with their chirality (CW/CCW) ? etc.

The chirality of the elliptical skyrmions is not determined by the direction of the in-plane field. We repeated the experiment in Fig. 3 for different field-tilting directions and found no difference (see following figure; added also to the Supplementary Material). The texture returns to the same state when the field is tilted back to the perpendicular direction (e).

(2) The most interesting point of the present thin plate system is that a variety of unique spin textures emerge depending on field-temperature history, which is very complicated to follow. So, I strongly recommend the authors to prepare a schematic phase diagram (B-T) of these spin textures as an additional Figure 5 or as Fig.4c, with short notes at the end of Discussion in the main text.

Thank you for this excellent suggestion. We have created a phase diagram (see below) but we emphasize that the phase diagram depends very much on the experimental protocol that is used. We have included this as a new figure in our revised manuscript. At temperatures of 300K and above we find antiskyrmions and at temperatures of 250 K and below we find elliptical-skyrmions as the predominant objects. We have added a discussion concerning this figure in the main text.

(3) As shown in Fig. 3, only CW elliptical-skyrmions were observed at 200K. By decreasing the temperature to 150K, coexistence of CCW and CW elliptical-skyrmions was observed as shown in Fig. 4b(i). Does it follow that only CCW elliptical-skyrmions are to be observed at lower temperatures than 150K? If the authors have already done their experiment at 90K or lower, I recommend the authors to include such results in their revised manuscript.

At 100K we only find isolated elliptical-skyrmions that directly transform to the helical state upon decreasing the magnetic field (see figure below). It is possible, that higher tilting angles are required in order to stabilize a periodic lattice of elliptical-skyrmions at these temperatures. However, in our experiment, the tilting angle is limited to 40° .

Therefore, to be able to answer this question, we investigated a thicker sample (280 nm) and found that a lattice of elliptical-skyrmions can form at 100 K. This is due to the increased relevance of the dipole-dipole interaction for thicker samples. Again, the skyrmions all have the same chirality (see Fig. below). There is no temperature dependence of the chirality.

Unfortunately, we cannot carry out measurements at lower temperatures in our TEM. However, since the chirality is apparently not determined by the temperature, we think this is not mandatory for our paper. We included a short discussion the two above figures in the manuscript.

We suspect that the origin for the one predominant type of chirality is strain. Even a small lattice deformation leads to the inequivalence of the DMI vectors along the $[100]$ and $[010]$ direction ($D_{[100]} \neq -D_{[010]}$). Consequently, one of the two skyrmion chiralities is energetically preferred even for minimal deformations. What brings us to this assumption, is that we observe a mixed state only in a different part of the sample (Fig. 5(ii)). Here, the lattice distortion may be smaller and both skyrmion chiralities can coexist since $D_{[100]} \approx -D_{[010]}$ (they are separated by an energy barrier). We added a short comment to the paper.

Revisions

(1) Title: "Discovery of elliptical-skyrmion chiral twins in an anti-skyrmion system"
Avoid "discovery". Choose an optimum title to be distinguished from the related work.

We have changed the title to "Elliptical Bloch skyrmion chiral twins in an anti-skyrmion system."

(2) Abstract, line 3: single "skyrmions" and arrays of "skyrmions" can be stabilized
I could not find any evidence that a single skyrmion is stabilized in Figs. 1-4, whereas sparse arrays of isolated skyrmions are observed in Figs. 1g,2a,3a. In addition, "skyrmion" is a vague expression. "A Bloch-type skyrmion with the skyrmion number of +1" is more precise.

We added the term "Bloch skyrmion" to the abstract and the main text. Also, we write now that skyrmions and antiskyrmions have opposite topological charges of ± 1 . Furthermore, we have changed "single skyrmions" to "sparse array of skyrmions"

(3) Abstract, line 5: "Dzyaloshinkii-Moriya" = Dzyaloshinskii-Moriya

Corrected. Thank you for spotting this typo.

(4) Abstract, line 7: "Here," Delete "Here". "Here" is already used in line 3.

We have adapted this suggestion.

(5) Abstract, line 12: "they are stabilized by long-range magnetostatic interactions"
"long-range magnetostatic interaction" is used only once in abstract. They use longrange dipole-dipole interaction" several times in the main text. Use a unified expression throughout the manuscript to avoid confusions if there is no special reason.

Corrected. Now we use "dipole-dipole interactions" throughout the whole manuscript.

(6) Page 2, Introduction:

Introduction is too short (256 words). Add more background to increase the volume conforming to the journal style.

We added a discussion on the predicted coexistence of skyrmions and anti-skyrmions in frustrated magnets and a few more details.

(7) Page 6, line 8-10: "When the initial Bloch skyrmion is chosen to have a counterclockwise chirality,..."

As this is illustrated in Fig. 4b(ii), add a reference at a proper position in the sentence.

We show the initial configuration in the supplement (newly added figure; will be shown in reply to comment 16). Figure 5 (former Fig. 4) in the manuscript shows the relaxed configuration instead. We added references to both figures accordingly in the text.

(8) Page 7, line 9: "Note that there exist reports of deformed or distorted skyrmions in several B20 materials."

Origin of the L-shaped deformation of skyrmions reported in ref. 20 is the anisotropic qvectors favoring [100] and [010] directions in $\text{Co}_8\text{Zn}_8\text{Mn}_4$. Other deformations as reported in refs. 21-24 were induced by confinement as correctly described in the present main text. Reference 20 must be described separately from references 21-24.

Thank you for this helpful comment. We changed the discussion of this reference.

(9) Page 8, Discussion:

Discussion is too short (1 paragraph). Add a short discussion on the phase diagram as a summary of the present work. Also, it would be interesting if they could add discussion on their additional experiments at a liquid nitrogen temperature (or even lower).

We have extended the discussion. Especially, we explain the dependence of our results on the experimental procedure, magnetic field and temperature, and refer to the phase diagram.

(10) Throughout the main text:

Describe the defocus values (distinguish underfocus or overfocus) and indicate the direction of out-of-plane bias-field by using a proper symbol (either a target mark or a crossed circle).

We have added a section on experimental details in "Methods" where we have included information on the defocus value. Also, we added the out-of-plane bias-field to Fig. 1a. It is the same in all the other figures.

(11) Figure 1g,2a,3a:

Describe whether these sparse isolated skyrmions remain even at vanishing field in the main text. I would like to know whether only a dense periodic lattice remains even after the bias-field is removed.

Only a dense lattice survives. If the field is decreased starting from a sparse array (see

following figure for antiskyrmions at 300 K), the helical phase becomes dominant at zero field. We have added a comment to the paper and included the figure in the Supplementary Material.

(12) Figures 1-4:

For readers' convenience, display the temperature as well as the strength of bias-field explicitly in all figures.

We followed this suggestion.

(13) Figures 1,2,4(vii),4(viii)

Poor contrast. Enhance contrast to improve visibility. (I will enhanced contrast and large are of e-Sk and aSk for referee 3)

We increased the contrast to the best of our capabilities. The image got a bit brighter but overall the contrast is limited by the experiment.

(14) Figure 1g

Display an arrow to show the direction of "stabilizing in-plane field [110]".

We added the [110] direction in Fig. 1g and added a comment to the figure captions. It is the same in all other figures.

(15) Figure 3a

I notice a periodic stripe running in the vertical direction. Add a note whether this is the non-magnetic contrast produced in FIB specimen preparation or magnetic origin.

Thank you for noticing it. It is the curtaining due to FIB preparation. We have added a comment in the new experimental methods section.

(16) Figure 4a(i),4a(ii), and 4b(ii)

Add the corresponding initial (circular?) states used to calculate the relaxed (elliptical) states as shown in 4a(i), 4a(ii), and 4b(ii) above each figure.

We thank you for this comment and understand the intention. We thought about adding these images ourselves before the initial submission but decided not to, since these states are not stable so they have no physical relevance. Therefore, we would still prefer to not show them in the manuscript. However, you are right that this technical detail is important.

Therefore, we hope that adding the following figure to the supplement is a good compromise. It visualizes how different circular skyrmionic objects relax in the presence of anisotropic DMI and dipole-dipole interactions. The three panels on the very right correspond to a(i), a(iv) and b(ii). The state a(ii) is achieved by changing the magnetic field strength.

(17) Figure 4b(i):

The figure contains an intermediate state with the skyrmion number of 0 (see Fig. 2b of the attached literature) in the near center of the field of view. Indicate this unique spin texture by an arrow and add a note in both main text and figure legend.

We have included this revision.

(18) Figure 4:

Mixed displays of simulations and experimental images in a figure are so confusing. Add proper labels (such as Sim. and Exp.) to help readers' comprehension. In addition, add temperatures and field strengths in appropriate positions of each figure.

We have added these labels.

(19) References, 29:

Complete the reference in accordance with the journal style. I could not check the relevance of this reference.

Thank you for this comment. The paper is not published, so we deleted the reference.

Reply to Referee #2

In this study, the authors continued their investigation at exploring the existence of novel spin textures that can be stabilized in the tetragonal inverse Heusler $\text{Mn}_{1.4}\text{Pt}_{0.9}\text{Pd}_{0.1}\text{Sn}$ with D_{2d} symmetry. In a previous work, they observed by Lorentz TEM for the first time the room temperature stabilization of a lattice of anti-skyrmions due to the presence of an anisotropic DM interaction in this material. Here, the authors unambiguously observe another topological phase at low temperature composed of elliptical skyrmions, in which the elongation direction determines the effective sense of rotation. In addition, at temperature higher than room temperature, they observe square-shaped skyrmions either isolated or in a lattice phase depending on the applied magnetic field.

Whereas the presence as well as the shape of the elongated skyrmions is corroborated by some micromagnetic simulations highlighting the importance of the long range dipole-dipole interaction, it would have been interesting to discuss more in detail the transition between circular and square skyrmionic textures at room temperature.

We carried out other experiments to come to the following conclusion: Experimentally, there seems to be an energy barrier between the round and square-shaped antiskyrmion state which we can clearly see in the experiment done at 300 K, shown in the following figure (added to the Supplementary Material). After stabilizing a round anti-skyrmion lattice (a), we decrease the field. A square antiskyrmion lattice forms (b). We increased the field again to the value where round antiskyrmions are typically stabilized, but without tilting the field in-plane. As a result, we still find square antiskyrmions (c). Finally, when tilting the field

in-plane and back, the states have transformed to round antiskyrmions.

Similarly, the authors explain the presence of only square skyrmions at higher than room temperature by invoking a decreased net magnetization at higher temperature. I believe that the authors should try to confirm this hypothesis by complementary experimental measurements completed by additional micromagnetic simulations. **Once done, I would recommend this work for publication in Nature Communications.**

We thank the referee for his/her positive comments about our results.

Our explanation is that at lower temperatures, the net magnetization is increased. Therefore, the relevance of the dipole-dipole interaction compared to the DMI is increased, which leads to an increased stability of skyrmions vs antiskyrmions (cf. also the newly added phase diagram) at low temperatures and an increased stability of antiskyrmions at elevated temperatures. We carried out additional experiments that are consistent with this explanation, as follows: besides changing the temperature, another way to change the net magnetization is by altering the sample thickness. Therefore, we investigated another sample at 100 K with a thickness of 280 nm (the sample we used in our original manuscript is 170 nm thick). For the thinner sample we barely find any skyrmions.

But for the thicker sample a nice array of elliptical-skyrmions forms.

This tells us that an increased net magnetization helps to stabilize skyrmions. We have added this figure and a discussion to the paper.

Reply to Referee #3

In this manuscript, Jena et al. report existence of elliptically deformed skyrmions in a Heusler compound with D_{2d} symmetry, which was previously found to host anti-skyrmions, by using Lorentz transmission electron microscopy technique. They argue that these nanometer-scale spin textures are stabilized by long range magneto static interaction by performing micromagnetic simulation that takes into account both DMI and dipole-dipole interaction.

I think that this paper contains novel and very interesting findings that **warrant publication in Nature Communication**, but before the publication, several points described below should be addressed to further polish the paper.

Thank you very much for your positive report. We appreciate your recommendation.

1) Authors apply a unique procedure for field application. What is the exact role or microscopic mechanism of the field tilting? How the in-plane field stabilizes antiskyrmions at high temperatures (300 -350 K) and skyrmions at low temperatures (150 K-250 K)? This point should be discussed.

We thank you for your question. Referee 1 had the same remark (remark 1) which we answered in detail. Here is again a short summary: we understand the in-plane tilting as a way to overcome the energy barrier between the topologically trivial phase and the skyrmion or antiskyrmion phase. Without tilting, the system remains in the helical phase, even though the energy is lower than in the topologically non-trivial phases. When an in-plane field is provided, the barrier is overcome and skyrmions or antiskyrmions can nucleate. We do not observe a fundamental difference in the role of the in-plane field for the formation of elliptical-skyrmions vs. antiskyrmions.

We have added several comments on pages 3 and 4. The behavior of the texture on field tilting along different directions can be seen in the following figure (also shown to Referee 1).

2) To me, Fig.4 (v) and Fig.4(vii) look very similar to each other. Resolution of Fig.4(vii) is not so good, and it may be possible that the objects are actually skyrmions, not antiskyrmions. Also, is the symmetry of the each texture compatible with the symmetry of hexagonal arrangements if the objects are anti-skyrmion?

It is indeed not so easy to see the difference looking only at these images, but especially when comparing the contrast with the simulated contrast given above, we think you can see that the antiskyrmion is more circular and the difference between the two bright and dark features are more pronounced. In order to better distinguish the two objects, we show here a lattice of these objects instead of only the isolated spin textures which we included in the revised supplementary information.

We hope you agree that there is a clear visual difference between these two objects. We have also attempted to increase the contrast and have included the improved figure in the main text.

Since the surrounding of skyrmions and antiskyrmion is collinear out-of-plane, these objects are in principle compatible with any lattice type and can exist even as individual objects, as we have shown at high fields. Arranging these objects in a hexagonal lattice is not problematic, geometrically speaking.

3) In this paper, authors state that at high temperatures at 300 K and 350 K, they observe antiskyrmions while at low temperatures (150-250 K), elongated skyrmions are observed. Are these results consistent with the phase diagram (Fig.4e) of the former Nature paper? In that figure, the whole phase diagram is dominated by antiskyrmions (although denoted as “skx”), and no skyrmion region is observed. What is the difference between the present and previous results?

The measurements in the nature paper were carried out in a different, thinner lamella although from the same crystal from which the lamellae in this manuscript were formed. Therefore, a different phase diagram is possible. Moreover, since the work was carried out for the Nature paper, we learned that the protocol (i.e. the field (tilting) and temperature history) that is used to carry out the LTEM is critically important in determining which phase is present. As we discuss in our current manuscript it is clear that elliptical-skyrmion and antiskyrmion phases are metastable. Thus, different experimental protocols lead to different magnetic textures. We believe that this is a very important result.

We have included a phase diagram, as one of the other referees proposed, in our revised manuscript (new Fig. 6 that is shown below). It is clear that this phase diagram is very different from the Nature paper and it even shows (the main point of our manuscript) an entirely new phase – the elliptical Bloch skyrmion. Still, the phase diagram in the Nature paper is correct using the protocols carried out in that paper on a thinner sample. We added a comment on the strong protocol-dependence of the results.

Reviewer #1 (Remarks to the Author):

2nd Review of "Elliptical Bloch skyrmion chiral twins in an anti-skyrmion system" by J. Jena et al. submitted to *Nature Communications*. Manuscript ID: NCOMMS_19_30742A

I (referee #1) have read the authors' responses to my comments, as well as those to other reviewers' comments on their previous manuscript. I am satisfied with their responses. In particular, the addition of a phase diagram (Fig. 6) has been successful to improve the manuscript more comprehensive. Now that their revised manuscript is much improved, I feel positive on their manuscript for a rapid publication in *Nature Communications*. However, I would like to suggest my final major and minor revisions to improve their revised manuscript further.

Major revisions

(1) I think the coexistence of the three kinds of spin textures in the same field-of-view as demonstrated in Supplementary Fig. 7 (Supplementary Fig. 3 in the previous manuscript) is a very important result. So, it is worth being displayed and explained in the main text. Move Supplementary Fig. 7 to Fig. 7 in the main text, and add a paragraph at the end of the Results section to explain the coexistence.

(2) Accordingly, indicate the two areas of the coexistence in the phase diagram (Fig. 6). The two areas correspond to the two conditions of Fig. 7a and b, respectively.

(3) Some readers might wonder how the elliptical skyrmion (Fig. 5(ii)) and square antiskyrmion (Fig. 5(iv)) appear in LTEM images when different defocus values are used. Add Supplementary Figure(s) demonstrating simulated LTEM images versus defocus values for the deformed (elliptical) skyrmion as shown in Fig. 5(ii) and the deformed (square-shaped) antiskyrmion as shown in Fig. 5(iv) (see the Extended Data Fig. 7 of their previous publication [1]).

(4) The authors' findings are many and intriguing. However, in my opinion, their presentations are poor. Figures 1-4 are especially boring; they are just simple arrays of LTEM images, lacking professional appearances. I recommend the authors to brush up their presentations of Figs. 1-4 to be more professional. How about adding schematic graphs explaining the experimental protocols just below the series of LTEM images how the LTEM images were acquired? As an example of such graphs, refer to Figs. 3a&4a in a previous publication [2] explaining how they changed temperature and field to acquire a series of diffraction patterns. In the present study, the authors must explain how they changed field and tilt angle to acquire their LTEM images at a fixed temperature in each figure, whereas the protocol is written in the main text.

(5) Fig. 6: As the density in the background is drawn in blue, symbols for the elliptical Bloch skyrmions (also painted in blue) are difficult to distinguish. Change the color of density so that all symbols are clearly distinguished.

[1] Nayak, A. K. et al. *Magnetic antiskyrmions above room temperature in tetragonal Heusler materials*. *Nature* 548, 561 (2017).

[2] K. Karube et al., Robust metastable skyrmions and their triangular–square lattice structural transition in a high-temperature chiral magnet. *Nature materials* **15**, 1237-1243 (2016).

Minor revisions

(1) Abstract, 1st sentence: "Anti-skyrmions and skyrmions are topological chiral spin textures" -> "Skyrmions and antiskyrmions are topologically different chiral spin textures" This is because (elliptical) skyrmion is the main topic of the manuscript.

(2) Abstract, 2nd sentence: "even sparse arrays of individual skyrmion"-> Better be deleted. I think the sparse array is not so essential for the present work. Moreover, "Only dense lattice survives" according to their response to my comment (11) in their response letter.

(3) Abstract, the last sentence: "The possibility of forming both anti-skyrmion and elliptical skyrmions, two chiral spin textures with opposite topological charges of ± 1 , in one material makes the family of D_{2d} materials exceptional."-> "The possibility of forming two distinct chiral spin textures with opposite topological charges of ± 1 in one material makes the family of D_{2d} materials exceptional."

(4) Page 3, line 51: "two distinct topological, chiral spin textures with opposite topological charges of" -> "two distinct chiral spin textures with opposite topological charges of"

(5) Page 3, line 57: "formed" -> "fabricated"

(6) Page 3, line 58: "FIB machining techniques" -> "FIB technique"

(7) Page 3, line 72: H_{sat} is not used in the following text. Can be deleted.

(8) Page 4, line 79-80: "to help to overcome the energy barrier to the reach a topologically non-trivial phase"->"to help to overcome the energy barrier"

(9) Page 10, line 246: "In summary," As they summarized in their rebuttal letter, their findings are many.

- (a) one of our most important discoveries is that the elliptical skyrmions are oriented along two distinct directions derived from the anisotropic DMI interaction.
- (b) the stabilization of lattices of square-shaped antiskyrmions, even at zero field
- (c) the stabilization of lattices of round antiskyrmions and even individual objects
- (d) the stabilization of lattices of elliptical-skyrmions, even at zero field, with two distinct elongation axes and chiralities
- (e) the coexistence of both types of elliptical skyrmions
- (f) the field-tilting dependence of the stability of elliptical-skyrmions
- (g) the switching between round and square-shaped antiskyrmions
- (h) the coexistence of skyrmions and antiskyrmions
- (i) the temperature, field and protocol dependence of their stability (phase diagram)
- (j) higher skyrmion stability for thicker samples
- (k) a physical explanation considering anisotropic DMI and dipole-dipole interactions
- (l) micromagnetic simulations that reproduce the experimental findings in agreement with our explanation

However, only (a),(d),(e),(f),(h), and (k) are included in summary. I think the large deformation of circular antiskyrmions at lower magnetic field appearing as

square-shaped antiskyrmions, as well as the deformation of elliptical skyrmions by decreasing magnetic field [(b)(c)(g)], is equally significant as the elliptical chiral twins. Include this finding at the end of the summary.

(10) Page 11, line 251: "the elliptical-skyrmions are chiral with a left-handed or right-handed chirality" -> "the elliptical skyrmions possess either a left-handed or right-handed chirality"

(11) Fig. 4, legend: Add explanations on ai,ii,iii & bi-viii.

(12) Fig. 6 (phase diagram): If the field value (vertical axis) corresponds to $\mu_0 H_{\text{tilt}}$ where H_{tilt} is defined in the protocol described in Page 4, line 78, use $\mu_0 H_{\text{tilt}}$ (mT) instead of $B(mT)$ as the axis label. Then, the definition of H_{tilt} is useful. Otherwise, it should be deleted, as the definition is useless.

(13) Throughout the manuscript: "anti-skyrmion"->"antiskyrmion" following the convention of the first report of antiskyrmion appeared on *Nature*.

End of 2nd Review

Reviewer #2 (Remarks to the Author):

The authors have properly answered my comments as well as the ones from the other referees. I also appreciate the changes done in the manuscript. I thus believe that this article can be now published in Nature Communications

Reviewer #3 (Remarks to the Author):

I have read the response from the authors. They revised the manuscript appropriately in response to my and other reviewers' comments although I am not fully satisfied with the response to my comment 1) as the explanation remains phenomenological level and not microscopic. I think that this interesting paper can now be accepted in Nature Communications and should be made available to general audience.

Reply to Referee #1

I (referee #1) have read the authors' responses to my comments, as well as those to other reviewers' comments on their previous manuscript. I am satisfied with their responses. In particular, the addition of a phase diagram (Fig. 6) has been successful to improve the manuscript more comprehensive. Now that their revised manuscript is much improved, I feel positive on their manuscript for a rapid publication in Nature Communications. However, I would like to suggest my final major and minor revisions to improve their revised manuscript further.

We are glad about the referee's recommendation and are very grateful for the final comments which we have all adapted to our manuscript, as discussed in the following.

(1) I think the coexistence of the three kinds of spin textures in the same field-of-view as demonstrated in Supplementary Fig. 7 (Supplementary Fig. 3 in the previous manuscript) is a very important result. So, it is worth being displayed and explained in the main text. Move Supplementary Fig. 7 to Fig. 7 in the main text, and add a paragraph at the end of the Results section to explain the coexistence.

We agree and added this figure (now Figure 6). Also, we added a short description to the end of the results section. A discussion / explanation was already included in the discussion section.

(2) Accordingly, indicate the two areas of the coexistence in the phase diagram (Fig. 6). The two areas correspond to the two conditions of Fig. 7a and b, respectively.

We added the data points corresponding to the coexistence at 268K to the phase diagram (now Figure 7).

(3) Some readers might wonder how the elliptical skyrmion (Fig. 5(ii)) and square antiskyrmion (Fig. 5(iv)) appear in LTEM images when different defocus values are used. Add Supplementary Figure(s) demonstrating simulated LTEM images versus defocus values for the deformed (elliptical) skyrmion as shown in Fig. 5(ii) and the deformed (square-shaped) antiskyrmion as shown in Fig. 5(iv) (see the Extended Data Fig. 7 of their previous publication [1]).

We follow this advice and have added a Supplementary Figure 5.

(4) The authors' findings are many and intriguing. However, in my opinion, their presentations are poor. Figures 1-4 are especially boring; they are just simple arrays of

LTEM images, lacking professional appearances. I recommend the authors to brush up their presentations of Figs. 1-4 to be more professional. How about adding schematic graphs explaining the experimental protocols just below the series of LTEM images how the LTEM images were acquired? As an example of such graphs, refer to Figs. 3a&4a in a previous publication [2] explaining how they changed temperature and field to acquire a series of diffraction patterns. In the present study, the authors must explain how they changed field and tilt angle to acquire their LTEM images at a fixed temperature in each figure, whereas the protocol is written in the main text.

We think that presenting the raw LTEM data is more honest than presenting colorful images from postprocessing. Still, we agree that adding a schematic image is very helpful. We added two of those to Fig. 1 in order to explain the experimental protocol that has been used. We did not add them to Figs. 2-4 because there the same protocol was used.

(5) Fig. 6: As the density in the background is drawn in blue, symbols for the elliptical Bloch skyrmions (also painted in blue) are difficult to distinguish. Change the color of density so that all symbols are clearly distinguished.

Thank you for this suggestion. We agree that the blue icons on a blue background were hard to see. Therefore, we changed the icon color to green.

Minor revision:

(1) Abstract, 1st sentence: "Anti-skyrmions and skyrmions are topological chiral spin textures" -> "Skyrmions and antiskyrmions are topologically different chiral spin textures" This is because (elliptical) skyrmion is the main topic of the manuscript.

(2) Abstract, 2nd sentence: "even sparse arrays of individual skyrmion"-> Better be deleted. I think the sparse array is not so essential for the present work. Moreover, "Only dense lattice survives" according to their response to my comment (11) in their response letter.

(3) Abstract, the last sentence: "The possibility of forming both anti-skyrmion and elliptical skyrmions, two chiral spin textures with opposite topological charges of ± 1 , in one material makes the family of D2d materials exceptional."-> "The possibility of forming two distinct chiral spin textures with opposite topological charges of ± 1 in one material makes the family of D2d materials exceptional."

(4) Page 3, line 51: "two distinct topological, chiral spin textures with opposite topological charges of" -> "two distinct chiral spin textures with opposite topological charges of"

(5) Page 3, line 57: "formed" -> "fabricated"

(6) Page 3, line 58: "FIB machining techniques" -> "FIB technique"

(7) Page 3, line 72: H_{sat} is not used in the following text. Can be deleted.

(8) Page 4, line 79-80: "to help to overcome the energy barrier to the reach a topologically non-trivial phase"->"to help to overcome the energy barrier"

(10) Page 11, line 251: "the elliptical-skyrmions are chiral with a left-handed or right-handed chirality" -> "the elliptical skyrmions possess either a left-handed or right-handed chirality"

(11) Fig. 4, legend: Add explanations on ai,ii,iii & bi-viii.

(13) Throughout the manuscript: "anti-skyrmion"->"antiskyrmion" following the convention of the first report of antiskyrmion appeared on Nature.

We have adapted all of these comments.

(12) Fig. 6 (phase diagram): If the field value (vertical axis) corresponds to $\mu_0 H_{\text{tilt}}$ where H_{tilt} is defined in the protocol described in Page 4, line 78, use $\mu_0 H_{\text{tilt}}$ (mT) instead of B(mT) as the axis label. Then, the definition of H_{tilt} is useful. Otherwise, it should be deleted, as the definition is useless.

It is not H_{tilt} . We updated the axis label to $\mu_0 H$ (mT) and deleted H_{tilt} from the text.

(9) Page 10, line 246: "In summary," As they summarized in their rebuttal letter, their findings are many.

- (a) one of our most important discoveries is that the elliptical skyrmions are oriented along two distinct directions derived from the anisotropic DMI interaction.
- (b) the stabilization of lattices of square-shaped antiskyrmions, even at zero field
- (c) the stabilization of lattices of round antiskyrmions and even individual objects
- (d) the stabilization of lattices of elliptical-skyrmions, even at zero field, with two distinct elongation axes and chiralities
- (e) the coexistence of both types of elliptical skyrmions
- (f) the field-tilting dependence of the stability of elliptical-skyrmions
- (g) the switching between round and square-shaped antiskyrmions
- (h) the coexistence of skyrmions and antiskyrmions
- (i) the temperature, field and protocol dependence of their stability (phase diagram)
- (j) higher skyrmion stability for thicker samples
- (k) a physical explanation considering anisotropic DMI and dipole-dipole interactions
- (l) micromagnetic simulations that reproduce the experimental findings in agreement with our explanation

However, only (a),(d),(e),(f),(h), and (k) are included in summary. I think the large deformation of circular antiskyrmions at lower magnetic field appearing as square-shaped antiskyrmions, as well as the deformation of elliptical skyrmions by decreasing magnetic field [(b)(c)(g)], is equally significant as the elliptical chiral twins. Include this finding at the end of the summary.

We agree that (b), (c) and (g) are also very important results and have added them to the discussion section.

Reply to Referee #2

The authors have properly answered my comments as well as the ones from the other referees. I also appreciate the changes done in the manuscript. I thus believe that this article can be now published in Nature Communications.

We thank the referee kindly for the recommendation.

Reply to Referee #3

I have read the response from the authors. They revised the manuscript appropriately in response to my and other reviewers' comments although I am not fully satisfied with the response to my comment 1) as the explanation remains phenomenological level and not microscopic. I think that this interesting paper can now be accepted in Nature Communications and should be made available to general audience.

We thank the referee kindly for the recommendation.